# Polymorphic Variants of Interleukin-13 R130Q and Interleukin-4 T589C in Children with and without Cow’s Milk Allergy

**DOI:** 10.3390/life12050612

**Published:** 2022-04-19

**Authors:** Oksana Matsyura, Lesya Besh, Olena Kens, Dana Kosorinová, Katarína Volkovová, Sandor G. Vari

**Affiliations:** 1Department of Pediatrics №2, Danylo Halytsky Lviv National Medical University, 79010 Lviv, Ukraine; lesya.besh@gmail.com; 2Institute of Hereditary Pathology of National Academy of Medical Sciences of Ukraine, 79000 Lviv, Ukraine; ovkens@gmail.com; 3Medical Faculty, Slovak Medical University in Bratislava, 833 03 Bratislava, Slovakia; dana.kosorinova@szu.sk (D.K.); katarina.volkovova@szu.sk (K.V.); 4International Research and Innovation in Medicine Program, Cedars-Sinai Medical Center, Los Angeles, CA 90048-4903, USA; vari@cshs.org

**Keywords:** nutritional allergy, cow’s milk, epigenetics, immunoglobulin E, gene polymorphism

## Abstract

Cow’s milk allergy (CMA) is one of the most frequent types of food allergy. The aim of the study was to investigate whether IL-13 R130Q and IL-4 T589C polymorphisms are associated with the risk of CMA in young Ukrainian children. A total of 120 children (age range: 1–3 years) participated in the study and were divided into two groups: CMA (*n* = 60) and healthy controls (CNT, *n* = 60). The CMA group had two subgroups: specific oral tolerance induction (SOTI, *n* = 30) and milk elimination diet (MED, *n* = 30). The CNT group had two subgroups: positive family history of allergy (+FHA, *n* = 24) and negative family history of allergy (−FHA, *n* = 36). In the CMA group, molecular genetic testing of CC, CT, and TT genotypes of single nucleotide IL-4 T589C gene polymorphisms showed significantly higher rates of the CC genotype compared to healthy controls (92.2% vs. 58.8%; *p* < 0.01). In the CMA group, molecular genetic testing of GG, GA, and AA genotypes of single nucleotide IL-13 R130Q gene polymorphisms showed significantly higher rates of GA and AA polymorphic locus genotypes compared to healthy controls (43.5% vs. 22.4%, *p* < 0.05 and 8.7% vs. 0%, *p* < 0.05, respectively). In future studies, the genotypic and allelic distribution of these polymorphic variants will be determined in children with CMA and healthy children.

## 1. Introduction

Human milk is an optimal source of food for infants due to its composition of essential nutrients, bioactive compounds, and immunological factors necessary for growth and optimal development [1]. It contains unreplaceable compounds for ideal postnatal development, since most infant formulas are lacking in regulatory factors [2,3,4].

The increase of allergic pathologies in the world is becoming epidemic [1]. The development of allergic diseases is caused by the complex interaction between genetic predisposition and epigenetic factors [3]. These genetic and epigenetic factors can lead to enzymatic methylation of deoxyribonucleic acid (DNA) and modification of its structure, and consequently its function [5]. In addition, epigenetic studies indicate that genes located on the 5q31-33 chromosome have a major influence on regulating basal serum immunoglobulin E (IgE) levels [6].

The increasing incidence of various food allergies, in particular, is due to many factors, including inadequate nutrition and harmful ecology (environmental pollution by industrial and household wastes) [4]. Epigenetics is considered to be one of the most promising areas of research for understanding the development of food allergies [7]. Understanding the epigenetic components that may underlie food allergies opens new perspectives because epigenetic components are reversible and can be modified or eliminated if necessary, in contrast to purely genetic components [8].

Cow’s milk allergy (CMA) is one of the most frequent types of food allergy in young children [2]. CMA is a complex disease characterized by a heterogenic phenotype, genetic aberrations, and different gene–environment interactions. Epigenetic factors are believed to play important roles in CMA because it is known that several environmental components such as maternal lifestyle, diet, hygiene, and stress are involved in the development of allergies. 

An allergic immune response targets various environmental allergens. It has been suggested that the tendency to develop a Th2 immune response is pronounced in atopic patients under the influence of genes and the microenvironment [9]. Subtypes of immune and inflammatory cells interact through cytokines. The roles of interleukin-13 (IL-13) and interleukin-4 (IL-4) genes have been implicated in atopic disease, producing an exaggerated IgE immune response to otherwise harmless substances [10]. 

Recently, several studies have found that polymorphisms in the IL-13 and IL-4 genes are often associated with an increased risk of allergic diseases, including atopic dermatitis, allergic rhinitis, and bronchial asthma [11,12]. At the same time, the influence of polymorphisms of these genes on the development of CMA in children remains unclear [13].

The aim of our study was to investigate whether polymorphisms of the IL-13 R130Q and IL-4 T589C genes are associated with the risk of developing CMA in young Ukrainian children. The study of these polymorphisms, which are located on the 5q31-33 chromosome, will facilitate further studies of the connection between certain mutations and the development of CMA. The selected polymorphisms are part of the well-known “The coding DNA sequence” database and are of interest to scientists around the world. 

## 2. Materials and Methods

### 2.1. Study Participants

In Lviv (Ukraine), 172 children (age range: 1–3 years) who were being treated with various dietary modification regimens were enrolled in the study. Ethical Committee approval was obtained from Danylo Halytsky Lviv National Medical University (22 May 2019 No. 5). The parents (or foster parents) of the selected children were informed and signed the written consent form for the child to participate in the study. 

Inclusion criteria for the study were: age 1 to 3 years, a positive skin test for cow’s milk (papules ≥ 3 mm by prick test), a milk-specific IgE level ≥ 0.35 IU/mL, and a positive oral food challenge (OFC) test for cow’s milk. Children with a history of anaphylaxis, severe comorbidities or autoimmune diseases, or individual contraindications to the OFC were excluded.

Of the 172 children who were originally enrolled in the study, 36 children withdrew and 16 children were excluded from the study due to protocol deviations.

A total of 120 children participated in the study and were divided into two groups: 60 children in the CMA group and 60 age-matched healthy children in the control group (CNT). The CMA group was randomized into two subgroups: the specific oral tolerance induction (SOTI, *n* = 30) subgroup and the milk elimination diet (MED, *n* = 30) subgroup. The CNT group was divided into two subgroups: positive family history of allergy (+FHA, *n* = 24) and negative family history of allergy (−FHA, *n* = 36). The laboratory and the genetic results were correlated.

### 2.2. OFC Test

The OFC test was performed at the Lviv City Children’s Allergology Center of the Communal Nonprofit Enterprise “City Children’s Clinical Hospital of Lviv,” according to the method developed by Professor Antonella Muraro of the University Hospital of Padua (Italy) [14]. 

Briefly, cow’s milk was given gradually to the child during the OFC test in portions every 15 to 30 min. If the child developed pathologic symptoms (skin or gastrointestinal), the test was stopped and he/she was then prescribed medications based on the nature of the symptoms. The child was observed at the Allergology Center for 2–6 h after the test. Later, the parents of the child were contacted by telephone about the child’s health at 24, 48, and 72 h after the test.

### 2.3. Measurement of Serum IgE Levels

Blood samples were collected to test total serum IgE and serum-specific IgE (sIgE) levels to individual cow’s milk allergens: casein—Bos d 8, α-lactalbumin—Bos d 4, and β-lactoglobulin—Bos d 5 (ALEX, MacroArrayDX, Vienna, Austria). 

### 2.4. IL-13 and IL-4

To assess the effectiveness of treatment, IL-13 and IL-4 levels were measured before the start of SOTI and at long-term follow-up (after 12 and 36 months). The levels of IL-13 and IL-4 were determined by sandwich ELISA (Thermo Fisher Scientific, Rockford, IL, USA). A target-specific antibody had been precoated in the wells of the supplied microplate. Samples, standards, or controls were then added into these wells and were bound to the immobilized (capture) antibody. The sandwich was formed by the addition of the second (detector) antibody. A substrate solution was added that reacted with the enzyme–antibody–target complex to produce a measurable signal. The intensity of this signal was directly proportional to the concentration of target present in the original specimen.

### 2.5. Genotyping

Comparative analysis of the rate of distribution of genotypes and alleles of the IL-13 R130Q and IL-4 T589C locus polymorphisms was performed on the data from the 60 children in the CMA group and 60 healthy children in the CNT group. To determine the genetic markers of allergy to cow’s milk proteins, a molecular genetic study of genotypes/alleles of polymorphic loci of IL-13 R130Q and IL-4 T589C genes was performed using polymerase chain reaction (PCR) and restriction fragment length polymorphism (RFLP) analyses. Genetically determined predisposition to CMA was evaluated based on polymorphism analysis of IL-13 R130Q (rs20541, Exo 4, G > A, Arg130Gln) and IL-4 T589C (rs2243250 = C-590T, promoter, T > C).

DNA was obtained from peripheral blood leukocytes of patients and isolated by a salting-out procedure. 

The IL-13 R130Q polymorphism was determined by gene amplification using two primers: forward 5′-CTTCCGTGAGGACTGAATGAGACGGTC-3′ and reverse 5′-GCAAATAATGATGCTTTCGAAGTTTCAGTGGA-3′ (Thermo Fisher Scientific). The amplification conditions were 4 min at 94 °C, 1 min at 69 °C, and 2 min at 72 °C, followed by 35 cycles of 30 s at 94 °C, 45 s at 67 °C, and 30 s at 72 °C, and then extension at 72 °C for 5 min. The PCR products were digested by addition of 0.25U NlaIV and incubated at 37 °C for 30 min. 

The IL-4 T589C polymorphism was determined by gene amplification using two primers: forward 5′-ACTACGCCTCACCTGANACG-3′ and reverse 5′-AGGTGTCGATTTGCAGTGAC-3′. The amplification conditions were 4 min at 94 °C, 1 min at 56 °C, and 1 min at 72 °C, followed by a final cycle of 72 °C for 7 min. The PCR products were digested by addition of BsmF1 and incubated at 65 °C for 6 h. 

Electrophoretic separation of PCR products and analysis of the RFLPs were performed in 2% agarose gel. Gels were stained with ethidium bromide and scanned on an ultraviolet transilluminator (BioRad Laboratories, Hercules, CA, USA). AmpliTaq DNA polymerases (Thermo Fischer Scientific, USA) were used. The samples were amplified in a thermocycler (BioRad Laboratories, Hercules, CA, USA).

### 2.6. Statistical Analysis

Statistical analysis of the study data was performed using the software packages RStudio v. 1.1.442 and R Commander v. 2.4-4 (serial number GPS-2158197-TGSV-68E7F, R Foundation for Statistical Computing, Vienna, Austria, 2020). Pearson’s criterion (*X^2^*) was used to compare the relative numbers of frequency and distribution and to assess the significance of the difference between them. The ratio of the frequency of the results in the CMA group to the frequency of the results in the CNT group was estimated using the value of the odds ratio (OR) with a 95% confidence interval (CI). A *p* value of <0.05 was considered statistically significant.

## 3. Results

In the CMA group, 60% of the SOTI subgroup and 46.7% of the MED subgroup were boys (*p* = 0.3). The mean ages at the start of treatment were 14.6 ± 3.86 months and 14.5 ± 3.18 months (*p* = 0.88), respectively, in the two subgroups.

In the CNT group, 56.10% of the +FHA subgroup and 50.0% of the −FHA subgroup were boys (*p* = 0.63). The mean ages at the start of the study were 15.63 ± 3.92 months and 17.42 ± 5.37 months, respectively, in the two subgroups (*p* = 0.16).

Serum levels of total IgE were 80.00 IU/mL in the SOTI subgroup and 44.5 IU/mL in the MED subgroup (*p* = 0.1008). Evaluation of the molecular profile allowed recording of the highest serum-specific IgE (sIgE) values against three major molecules. The sIgE values in the SOTI and MED subgroups, respectively, were 2.10 and 2.00 kUa/L (*p* = 0.1015) to Bos d 4 (α-lactalbumin), 1.90 and 1.55 kUa/L (*p* = 0.2034) to Bos d 8 (casein), and 0.85 and 1.60 kUa/L (*p* = 0.4768) to Bos d 5 (β-lactoglobulin). 

At the beginning of the study, mean IL-13 levels were higher in children with CMA (8.893 ± 0.684 pg/mL in SOTI subgroup, 8.235 ± 1.142 pg/mL in MED subgroup) compared to healthy controls (3.635 ± 0.313 pg/mL in +FHA subgroup, 3.781 ± 0.409 pg/mL in −FHA subgroup). In children who received SOTI, the IL-13 level significantly decreased over time (*p* < 0.05 after 12 months and *p* < 0.01 after 36 months).

Mean IL-4 levels at the beginning of the study were higher in children with CMA (1.209 ± 0.080 pg/mL in SOTI subgroup, 1.029 ± 0.069 pg/mL in MED subgroup) compared to healthy controls (0.429 ± 0.055 pg/mL in +FHA subgroup, 0.397 ± 0.032 pg/mL in −FHA subgroup). 

The distribution rates of genotypes and alleles of the IL-13 R130Q and IL-4 T589C locus polymorphisms were studied in 104 and 102 children, respectively (Table 1). DNA sequencing at the IL-13 R130Q locus could not be performed for technical reasons in 16 children, and could not be performed at the IL-4 T589C locus in 18 children. The ORs for the development of CMA were also calculated. 

Children with CMA had a significantly higher rate of GA and AA genotypes of the IL-13 R130Q locus polymorphism compared to healthy controls (GA: 43.5% vs. 22.4%, *p* < 0.05; AA: 8.7% vs. 0%, *p* < 0.05) (Table 1, Figure 1). There was a significantly higher rate of the GG genotype of the IL-13 R130Q locus polymorphism in healthy children of the control group compared to children with CMA (77.6% vs. 47.8%, *p* < 0.01), which suggests a protective effect of the GG genotype regarding the risk of developing CMA in young children.

Analysis of the distribution of alleles of the IL-13 R130Q polymorphism showed that the G allele was significantly more common in healthy children of the control group compared to children with CMA (88.8% vs. 69.6%, *p* < 0.01) (Table 1, Figure 1). Conversely, the A allele was significantly more common in children with CMA than in healthy controls (30.4% vs. 11.2%, *p* < 0.01). Thus, the A allele was over three times more common in children with CMA (OR = 3.47, 95% CI: 1.67–7.18; *p* < 0.01).

Analysis of molecular genetic testing of CC, CT, and TT genotypes of the IL-4 T589C single nucleotide polymorphism showed a significantly higher rate of the CC genotype of the IL-4 T589C locus polymorphism in the CMA group compared to healthy controls (92.2% vs. 58.8%, *p* < 0.01) (Table 1, Figure 2). Significantly higher rates of CT and TT genotypes of the IL-4 T589C locus polymorphism were found in healthy controls compared to the CMA group (CT: 25.5% vs. 7.8%, *p* < 0.05; TT: 15.7% vs. 0%, *p* < 0.01). 

These findings suggest protective properties of the CT and TT genotypes of the IL-4 T589C locus polymorphism in relation to the risk of developing CMA in young children.

Analysis of the distribution of alleles of the IL-4 T589C polymorphism showed that the T allele was significantly more common in the control group of healthy children compared to children in the CMA group (28.4% vs. 3.9%, *p* < 0.01) (Table 1, Figure 2). Conversely, the C allele was significantly more common in the CMA group than in the control group (96.1% vs. 71.6%, *p* < 0.01).

For children who live in urban areas, there was a significantly higher rate of the GG genotype of the IL-13 R130Q locus polymorphism in the CNT group of healthy children compared to the CMA group (75.5% vs. 46.7%, *p* < 0.05) (Table 2). There was a significantly higher rate of the GA genotype of the IL-13 R130Q locus polymorphism in urban children with CMA compared to healthy controls who live in urban areas (53.3% vs. 24.5%, *p* < 0.05). 

The analysis of molecular genetic testing of the CC genotype of the IL-4 T589C locus polymorphism showed a significantly higher rate of this genotype in urban children with CMA compared to healthy controls who live in urban areas (92.3% vs. 55.3%, *p* < 0.05) (Table 2). Rates of the CT and TT genotypes were not significantly different between the two groups.

For children who live in rural areas, there was a significantly higher rate of the GG genotype of the IL-13 R130Q gene polymorphism in the control group of healthy children than in children with CMA (100.0% vs. 48.4%, *p* < 0.05) (Table 3). Rates of the other genotypes of the IL-13 R130Q and IL-4 T589C gene polymorphisms were not significantly different between the two groups.

Analysis of the data from children with CMA indicates that the distribution rates of genotypes of the IL-13 R130Q and IL-4 T589C gene polymorphisms were not significantly different whether the children lived in urban or rural areas (Table 4). Sample sizes in the urban subgroups (i.e., IL-13 or IL-4) were comparable, as were those for the rural subgroups. 

Analysis of the data from the control group of healthy children indicates that the distribution rates of genotypes of the IL-13 R130Q and IL-4 T589C gene polymorphisms were not significantly different whether the children lived in urban or rural areas (Table 5). Sample sizes in the urban subgroups (i.e., IL-13 or IL-4) were comparable, as were those for the rural subgroups. 

Analysis of molecular genetic testing of the GG, GA, and AA genotypes of the single nucleotide IL-13 R130Q gene polymorphism showed that the CMA group had a significantly higher rate of GA and AA polymorphic locus genotypes compared to healthy controls (43.5% vs. 22.4%; *p* < 0.5 and 8.7% vs. 0%; *p* < 0.05, respectively). Analysis of the allelic distribution of the IL-13 R130Q gene polymorphism indicated that the A allele was more common in the group of children with CMA than in their healthy peers (30.4% vs. 11.2%; *p* < 0.01).

Analysis of molecular genetic testing of the CC, CT, and TT genotypes of the single nucleotide IL-4 T589C gene polymorphism showed a significantly higher rate of the CC genotype in the CMA group compared with healthy controls (92.2% vs. 58.8%; *p* < 0.01). Significantly higher frequencies of the CT and TT genotypes of the IL-4 T589C locus polymorphism were recorded in healthy controls compared with the CMA group (25.5% vs. 7.8%; *p* < 0.05 and 15.7% vs. 0%; *p* < 0.01, respectively). The analysis of the distribution of alleles of the IL-4 T589C gene polymorphism indicated that the T allele was significantly more common in healthy children than in children with CMA (28.4% vs. 3.9%; *p* < 0.01).

## 4. Discussion

Food allergy is an important medical and social problem in pediatric science and practice. In the ranking of food sensitization in young children, CMA is in the top position, mainly due to IgE-dependent mechanisms [2]. According to our data, not all existing possibilities are used in the diagnosis and treatment of IgE-dependent CMA; in particular, the SOTI method is used to a very limited extent [15]. Therefore, improving the effectiveness of early diagnosis and treatment of IgE-dependent CMA by developing and implementing an optimized set of diagnostic and therapeutic measures is a topical issue for modern pediatrics.

The development of food allergies in children depends on the effects of genetic factors, the nature of nutrition, environmental issues, social factors, and external care. From the standpoint of clinical genetics, allergy is treated as a multifactorial polygenic disease. To date, it has been established that not a single nosological entity of food allergies is inherited, but genetic factors are a predisposition to their development [3]. There are data on several hundred genetic markers associated with predisposition to allergic diseases or their individual phenotypic manifestations [16].

In recent years, much research has been conducted studying the genetic markers of allergic pathology. Research in recent years has identified five DNA loci that are most common in children with CMA [17]. In particular, the first chromosome contains the filaggrin gene, which is involved in the formation of the protective epithelial barrier. Mutations in this gene explain the mechanisms of inflammatory skin diseases and reduced immune response, which are more common in children with food allergies and atopic dermatitis [18].

Today, it has been proven that the long arm of the fifth chromosome regulates cytokine genes involved in the allergic process, namely IL-4, IL-5, IL-6, IL-9, IL-12, and IL-13 [8]. Cytokines are important mediators of intercellular interactions that regulate the immune response as well as the cell cycle, and are involved in numerous physiological and pathological processes. Of course, the leading role is played by the IL-4 and IL-13 genes involved in the development of inflammatory reactions during the exacerbation of allergies, including the work of regulatory T-cells of mucous membranes and skin [19].

The eleventh chromosome may contain a single nucleotide substitution, which, according to the literature, is much more common in children with CMA, due to IgE-independent mechanisms [20].

There is evidence that the short arm of the eighteenth chromosome contains a region of genes that synthesize Serpin family B number 10 (SERPIN B 10) proteins associated with food allergies. The two most common single nucleotide polymorphisms are located in the regulatory regions of genes that regulate the work of type 2 T helpers (Th2). It should be noted that this protein is currently being actively studied, as its function remains unclear. SERPIN proteins are of great interest to scientists because they suggest that studying the mechanism of their expression on the skin and mucous membranes may shed new light on the pathogenesis and understanding of food allergies [21].

There has been much discussion in the past about the association of human leukocyte antigens on chromosome 6 with food allergies, but to date, this information has been confirmed only in the case of allergies to peanuts [22].

The study of genetic factors in the development of food allergies remains an urgent problem today. Various genetic polymorphisms are actively analyzed, as they have the greatest impact on the synthesis of general and specific IgE, the production of proinflammatory cytokines, and the expression of IgE receptors on immunocompetent cells [23].

IL-4 and IL-13 are related cytokines that regulate many aspects of allergic inflammation and atopy in general [24]. The IL-4 gene is often referred to as the “critical inflammatory cytokine” gene, which activates humoral immunity and enhances immunoglobulin E production. The IL-13 gene is located on the 5q31 chromosome, which also encodes other factors involved in allergy mechanisms such as IgE, IL-4, IL-3, and IL-5 [25]. IL-13 is known to cause alternative activation of macrophages involved in the activation of Th2 reactions [26]. To date, polymorphic variants of cytokine genes are widely studied by scientists in the context of risk factors associated with diseases such as bronchial asthma, allergic rhinitis, glioma, hepatocellular carcinoma, idiopathic nephrotic syndrome, and orthodontic pathology [27,28,29].

Thus, a meta-analysis conducted by Chinese scientists showed an association between the IL-13 rs20541 (R130Q) gene polymorphism and susceptibility to glioma, as the GA genotype was much more common in glioma patients [29]. Al Rushood et al. analyzed the relationship of the IL-13 R130Q (rs20541) polymorphism to the predisposition of children to idiopathic nephrotic syndrome and found that the IL-13 RQ genotype was significantly more common in children with high sensitivity to steroid therapy than in children with steroid resistance [27]. However, they did not find an association between the IL-13 R130Q polymorphism (rs20541) and an increased risk of developing idiopathic nephrotic syndrome. There was no significant difference between the IL-13 R130Q (rs20541) polymorphism and the risk of asthma in a study by Iranian scientists [30]. They explained these results by stating that polymorphic variants of the IL-13 gene are not the only direct cause of diseases such as asthma.

Interesting results were obtained by Yadav et al., who studied the predisposition to allergic rhinitis [31]. A molecular genetic study of the IL-13 R130Q (rs20541) locus polymorphism showed a significantly increased rate of the GA genotype and A allele in patients with allergic rhinitis, in contrast to healthy people in the control group. However, homozygotes of the AA genotype showed no association with allergic rhinitis. The distribution of genotypes of the IL-4 T589C locus polymorphism showed that the rate of the CC genotype and C allele was statistically greater in patients with allergic rhinitis, in contrast to healthy controls. There was no significant difference between the genotypes of the IL-13 R130Q (rs20541) locus polymorphism in the study of predisposition to asthma. However, it was proven that the A allele is associated with a high chance of developing asthma.

Shumna et al. analyzed the relationship of the IL-4 T589C polymorphism with bronchial asthma, allergic rhinitis, and orthodontic pathology and concluded that the CC genotype of the IL-4 gene is associated with bronchial asthma (OR = 4.31; 95% CI = 1.63–11.36; *p* = 0.002) and allergic rhinitis (OR = 4.32; 95% CI = 1.04–7.81; *p* = 0.04) [32]. The TT genotype showed a predisposition to the development of orthodontic pathology. 

A meta-analysis that included 55 studies conducted by Kousha et al. showed that the CT genotype of the IL-4 T589C locus polymorphism was associated with an increased risk of asthma in the general population, regardless of ethnicity [17]. It was also found that the AA genotype of the IL-13 R130Q (rs20541) locus polymorphism was significantly more common in children with asthma than in healthy subjects. The polymorphic variant of IL-4 T589C was studied in the context of a possible connection with the development of hepatocellular carcinoma. The results of the analysis showed that the CT genotype of the IL-4 T589C locus polymorphism was significantly more common in the group of patients with hepatocellular carcinoma compared with the group of patients with liver cirrhosis and the group of healthy individuals. Moreover, the homozygous TT genotype was more common in the group of patients with hepatocellular carcinoma (24.0%) compared with liver cirrhosis (5.0%) and controls (3.3%). Carriers of the homozygous TT genotype have been shown to have increased risk for hepatocellular carcinoma by 10-fold compared to healthy subjects and by 6.33-fold compared to the group of patients with cirrhosis (*p* = 0.018 and 0.016, respectively).

Our findings are concordant with the results of many scientists around the world who have studied similar polymorphic gene variants as the focus of risk for atopic diseases, which indicates that our research aims are valid. Because CMA manifests in the first months of a child’s life, it is a side effect of food involving immune mechanisms and is a key step in the process of the atopic progression. Unfortunately, the problem of CMA is not well-studied both in Ukraine and abroad, although its importance in today’s world continues to grow.

Given the available data from the literature and the results of our own research, it can be argued that the impact of various factors on the child’s body can significantly change the profile of gene expression, thereby determining the characteristics of individual responses. Nucleotide polymorphism has become a new approach to the identification and localization of genetic determinants of atopy. Genetic predisposition to CMA is probably caused by IL-13 R130Q and IL-4 T589C polymorphisms involved in the mechanism of regulation of allergic reactions. The identification of these patterns has both diagnostic and prognostic value for the management of CMA in children.

To conduct immunological monitoring of the effectiveness of different methods of dietary modification, it is reasonable to study the changes over time of IL-13 and IL-4 in children with CMA, as the choice of cytokines should be based on understanding the functional activity and role of different classes of immunocompetent cells, type 1 T helpers (Th1), and Th2. Under normal conditions, activation of Th1 leads to a normal immune response, while stimulation of Th2 leads to the development of allergies. Th1 cells (via INF-γ, IL-2, IL-12) stimulate the cellular response by activating macrophages, natural killers, and cytotoxic lymphocytes and switch the synthesis from IgM to IgG and IgA; Th2 cells (via IL-4, IL-5, IL-6, IL-10, IL-13) affect eosinophils and B-lymphocytes and switch the synthesis from IgM to IgG, IgA, and IgE [33].

The study of the changes over time of IL-13 and IL-4 in the group of children with CMA and healthy people is considered important for understanding the risks of developing the disease and assessing therapeutic effectiveness. In addition, the environment, in turn, plays an important role in the development of sensitization to different groups of allergens (household, epidermal, fungal, pollen) [9,29].

The diagnosis of CMA can be quite simple if the child has persistent or recurrent symptoms that are associated with the intake of certain foods [16]. However, it is often complicated because the clinical manifestations observed in children can have various causes [34]. In addition, the symptoms themselves tend to change with the age of the child, and they can be caused by different foods; this requires a comprehensive diagnostic approach. Diagnosis of CMA should begin with a detailed family allergy history, in particular taking into account information about relatives of I and II levels of kinship [35]. Paraclinical methods include pretests with food allergens as well as specific IgE to identify extracts and molecules (components) of allergens [2]. Important components of milk are casein (Bos d 8), β-lactoglobulin (Bos d 5), α-lactalbumin (Bos d 4), and bovine serum albumin (Bos d 6). Knowing the characteristics of individual components of milk is important for assessing the risk of anaphylaxis and choosing further diagnostic and therapeutic approaches [20].

In recent years, the treatment of IgE-mediated CMA, in particular through dietary modification, has been actively discussed. Until recently, it was believed that a strict elimination diet should be unequivocally prescribed when a food allergy is detected in infants [36]. However, it should be noted that an elimination diet for long-term use poses risks of impaired nutrition and reduced quality of life for children and their parents and often does not solve the problem of allergies [37]. At the same time, today there is a unique opportunity to carry out targeted dietary modification through reasonable influence on immune mechanisms based on the use of the SOTI method [2]. The data obtained to date have shown that the use of this therapeutic method can change the composition of cytokine markers, as well as reduce the sensitivity of the skin, mucous membranes of the respiratory system, and digestive tract to allergen exposure, and can slow and even stop the progression of the “atopic march”. This method of treatment looks to be the most promising and requires in-depth study and wider implementation in pediatric practice. However, SOTI requires the development of special skills in both the doctor and the patient and his/her family members, and this is probably why it is very rarely used in practical pediatrics in Ukraine.

Recently, there has been growing interest in SOTI, the only method of medical intervention that can affect the natural course of an allergic disease and can not only effectively alleviate symptoms, but can also prevent disease progression [21,34]. The results of global research summarize and emphasize the considerations that long-term strategies for food allergy management should include a comprehensive diagnosis, a personalized approach to treatment, and nutritional status monitoring, as well as educational and non-pharmacological activities [38,39].

## 5. Conclusions

The presence of the GA genotype of the IL-13 R130Q locus polymorphism significantly increases a child’s risk of developing CMA (*p* < 0.05). A significantly higher rate of the GG genotype of the IL-13 R130Q locus polymorphism was found in healthy children compared to children with CMA (*p* < 0.01), which suggests a protective effect of the GG genotype in relation to the risk of CMA in young children. The G allele was significantly more common in healthy children (*p* < 0.01), and the A allele was more common in the group of children with CMA (*p* < 0.01).

The presence of the CC genotype of the IL-4 T589C locus polymorphism significantly increases the risk of CMA in a child (*p* < 0.01). Significantly higher rates of the CT (*p* < 0.05) and TT (*p* < 0.01) genotypes of the IL-4 T589C locus polymorphism were observed in healthy children. These results suggest a protective role of the CC and CT genotypes of the IL-4 T589C locus polymorphism in relation to the risk of developing CMA in young children. The C allele was significantly more common in the group of children with CMA (*p* < 0.01). The T allele was significantly more common in healthy children of the control group (*p* < 0.01).

For children living in urban areas, a significantly higher rate of the GG genotype of the IL-13 R130Q locus polymorphism was found in healthy children of the control group compared to children in the CMA group (*p* < 0.05). In contrast, in urban areas, a significantly higher rate of the GA genotype of the IL-13 R130Q locus polymorphism was found in children of the CMA group than in children of the control group (*p* < 0.05). Analysis of molecular genetic testing of the CC genotype of the IL-4 T589C locus polymorphism showed a significantly higher rate of this genotype in urban children of the CMA group compared to healthy controls who live in urban areas (*p* < 0.05). 

## Figures and Tables

**Figure 1 life-12-00612-f001:**
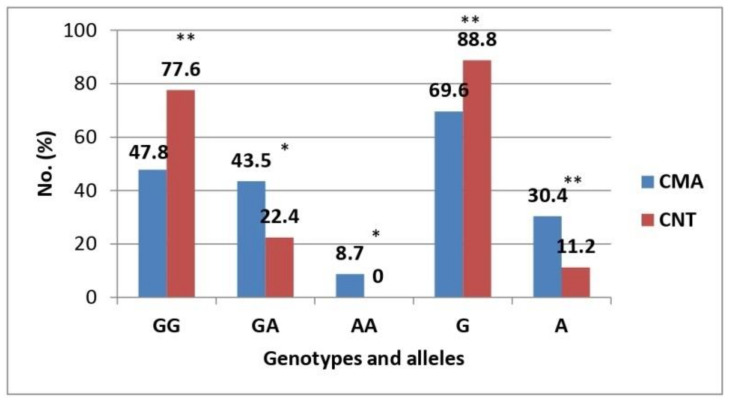
Distribution rates of genotypes and alleles of the IL-13 R130Q locus polymorphism. CMA, cow’s milk allergy group; CNT, control group. Note: *—*p* < 0.05; **—*p* < 0.01.

**Figure 2 life-12-00612-f002:**
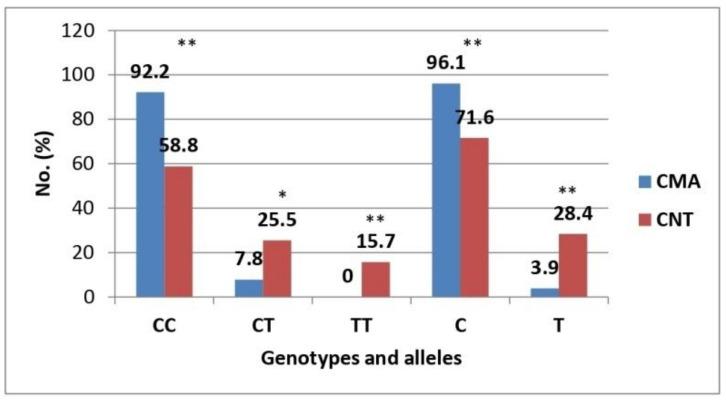
Distribution rates of genotypes and alleles of the IL-4 T589C locus polymorphism. CMA, cow’s milk allergy group; CNT, control group. Note: *—*p* < 0.05; **—*p* < 0.01.

**Table 1 life-12-00612-t001:** Distribution rates of genotypes and alleles of IL-13 R130Q and IL-4 T589C locus polymorphisms.

Detection of Genotypes and Alleles in Groups	CMA No. (%)	CNT No. (%)	X^2^	*p*	OR (95% CI)
*IL-13* R130Q genotypes	*n* = 46	*n* = 58			
GG	22 (47.8)	45 (77.6)	9.91 **	<0.01	0.26 (0.11–0.62)
GA	20 (43.5)	13 (22.4)	5.25 *	<0.05	2.66 (1.14–6.22)
AA	4 (8.7)	0 (0)	5.25 *	<0.05	0.000 (0.000-n/a)
Alleles	*n* = 92	*n* = 116			
G	64 (69.6)	103(88.8)	11.99 **	<0.01	0.29 (0.14–0.6)
A	28 (30.4)	13 (11.2)	11.99 **	<0.01	3.47 (1.67–7.18)
*IL-4* T589C genotypes	*n* = 51	*n* = 51			
CC	47 (92.2)	30 (58.8)	15.31 **	<0.01	8.23 (2.57–26.32)
CT	4 (7.8)	13 (25.5)	5.72 *	<0.05	0.25 (0.07–0.83)
TT	0 (0)	8 (15.7)	8.68 **	<0.01	0.000 (0.000-n/a)
Alleles	*n* = 102	*n* = 102			
C	98 (96.1)	73 (71.6)	22.59 **	<0.01	9.73 (3.28–28.9)
T	4 (3.9)	29 (28.4)	22.59 **	<0.01	0.1 (0.03–0.31)

CI, confidence interval; CMA, cow’s milk allergy group; CNT, control group; IL-4, interleukin-4; IL-13, interleukin-13; n/a, not applicable; OR, odds ratio. Note: *—*p* < 0.05; **—*p* < 0.01.

**Table 2 life-12-00612-t002:** Distribution rates of genotypes of IL-13 R130Q and IL-4 T589C locus polymorphisms in children who live in urban areas.

Detection of Genotypes in Groups	CMA (Urban) No. (%)	CNT (Urban) No. (%)	X^2^	*p*	OR (95% CI)
*IL-13* R130Q genotypes	*n* = 15	*n* = 53			
GG	7 (46.7)	40 (75.5)	4.54 *	<0.05	0.28 (0.09–0.94)
GA	8 (53.3)	13 (24.5)	4.54 *	<0.05	3.52 (1.07–11.58)
AA	0 (0)	0 (0)	-	-	-
*IL-4* T589C genotypes	*n* = 13	*n* = 47			
CC	12 (92.3)	26 (55.3)	6.0 *	<0.05	9.69 (1.16–80.71)
CT	1 (7.7)	13 (27.7)	2.27	>0.05	0.22 (0.03–1.85)
TT	0 (0)	8 (17.0)	2.55	>0.05	0.000 (0.000-n/a)

CI, confidence interval; CMA, cow’s milk allergy group; CNT, control group; IL-4, interleukin-4; IL-13, interleukin-13; n/a, not applicable; OR, odds ratio. Note: *—*p* < 0.05.

**Table 3 life-12-00612-t003:** Distribution rates of genotypes of IL-13 R130Q and IL-4 T589C locus polymorphisms in children who live in rural areas.

Detection of Genotypes in Groups	CMA (Rural) No. (%)	CNT (Rural) No. (%)	X^2^	*p*	OR (95% CI)
*IL-13* R130Q genotypes	*n* = 31	*n* = 5			
GG	15 (48.4)	5 (100.0)	4.65 *	<0.05	0.000 (0.000-n/a)
GA	12 (38.7)	0 (0)	2.9	>0.05	n/f (n/a-n/f)
AA	4 (12.9)	0 (0)	0.73	>0.05	n/f (n/a-n/f)
*IL-4* T589C genotypes	*n* = 38	*n* = 4			
CC	35 (92.1)	4 (100.0)	0.34	>0.05	0.000 (0.000-n/a)
CT	3 (7.9)	0 (0)	0.34	>0.05	n/f (n/a-n/f)
TT	0 (0)	0 (0)	-	-	-

CI, confidence interval; CMA, cow’s milk allergy group; CNT, control group; IL-4, interleukin-4; IL-13, interleukin-13; n/a, not applicable; n/f, not found; OR, odds ratio. Note: *—*p* < 0.05.

**Table 4 life-12-00612-t004:** Distribution rates of genotypes of the IL-13 R130Q and IL-4 T589C gene polymorphisms in children with cow’s milk allergy (CMA) from urban and rural areas.

Detection of Genotypes in Groups	CMA (Urban) No. (%)	CMA (Rural) No. (%)	X^2^	*p*	OR (95% CI)
*IL-13* R130Q genotypes	*n* = 15	*n* = 31			
GG	7 (46.7)	15 (48.4)	0.01	>0.05	0.93 (0.27–3.21)
GA	8 (53.3)	12 (38.7)	0.88	>0.05	1.81 (0.52–6.29)
AA	0 (0)	4 (12.9)	2.12	>0.05	0.000 (0.000-n/a)
*IL-4* T589C genotypes	*n* = 13	*n* = 38			
CC	12 (92.3)	35 (92.1)	0	>0.05	1.03 (0.1–10.85)
CT	1 (7.7)	3 (7.9)	0	>0.05	0.97 (0.09–10.26)
TT	0 (0)	0 (0)	-	-	-

CI, confidence interval; IL-4, interleukin-4; IL-13, interleukin-13; n/a, not applicable; OR, odds ratio.

**Table 5 life-12-00612-t005:** Distribution rates of genotypes of the IL-13 R130Q and IL-4 T589C gene polymorphisms in the control group (CNT) of healthy children from urban and rural areas.

Detection of Genotypes in Groups	CNT (Urban) No. (%)	CNT (Rural) No. (%)	X^2^	*p*	OR (95% CI)
*IL-13* R130Q genotypes	*n* = 53	*n* = 5			
GG	40 (75.5)	5 (100.0)	1.58	>0.05	0.000 (0.000-n/a)
GA	13 (24.5)	0 (0)	1.58	>0.05	n/f (n/a-n/f)
AA	0 (0)	0 (0)	-	-	-
*IL-4* T589C genotypes	*n* = 47	*n* = 4			
CC	26 (55.3)	4 (100.0)	3.04	>0.05	0.000 (0.000-n/a)
CT	13 (27.7)	0 (0)	1.48	>0.05	n/f (n/a-n/f)
TT	8 (17.0)	0 (0)	0.81	>0.05	n/f (n/a-n/f)

CI, confidence interval; IL-4, interleukin-4; IL-13, interleukin-13; n/a, not applicable; n/f, not found; OR, odds ratio.

## Data Availability

The data presented in this study are available on request from the corresponding author.

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
