# Peer review of "Polymorphic Variants of Interleukin-13 R130Q and Interleukin-4 T589C in Children with and without Cow’s Milk Allergy"

_life, 2022, doi:10.3390/life12050612_

Round 1

Reviewer 1 Report

The authors of this work investigate the association of IL-13 R130Q and IL-4 T589C polymorphisms with the risk of cow’s milk allergy in children. They found that IL-4 T589C gene polymorphisms showed significantly higher rates of the CC genotype and IL-13 R130Q gene polymorphisms showed significantly higher rates of GA and AA polymorphic locus genotypes in CMA children compared to healthy controls.

Major revisions are needed:

  1. The Indroduction Section is not well structured but chaotic. Why talk about epigenetics?
  2. Which polymorphisms do the references refer to 11, 12? Are they the same SNPs analyzed by you?
  3. Why did you choose these 2 polymorphisms?
  4. In the Methods Section, the Genotyping musty be better described (Linkage disequilibrium analysis)
  5. The MAF of these 2 SNPs could be reported
  6. The figures are too small and unclear
  7. Is there a correlation between serum levels of IL-4 and IL-5 and their respective polymorphisms?
  8. What is known about the function of these 2 polymorphisms on the gene expression?

Author Response

Response to Reviewer 1 

Dear Reviewer 1! Thank You for Your questions and recommendations!

1. The Indroduction Section is not well structured but chaotic. Why talk about epigenetics?

Response: We thank Reviewer 1 for recommending these improvements to the Introduction section. We have restructured and revised the Introduction to address these concerns. We discuss epigenetics for several reasons: 1) the development of allergic diseases is caused by the complex interaction between genetic predisposition and epigenetic factors; 2) epigenetic studies indicate that genes located on the 5q31-33 chromosome have a major influence on regulating basal serum immunoglobulin E (IgE) levels; and 3) epigenetics is considered to be one of the most promising areas of research for understanding the development of food allergies.

2. Which polymorphisms do the references refer to 11, 12? Are they the same SNPs analyzed by you?

Response: We thank Reviewer 1 for these interesting questions. The authors of reference #11 did not specify the SNPs in the article.  At the end of the Methods section, the authors state: “Genotypes of the used SNPs, … are available on reasonable request and, according to the Dutch privacy law, only after a data transfer agreement.” The authors of reference #12 analyzed four polymorphisms, two of which were the same ones we analyzed (IL-13 R130Q, IL-4 T589C).

3. Why did you choose these 2 polymorphisms?

Response: We thank Reviewer 1 for this question. We chose these two  polymorphisms because they are associated with the risk of developing allergic diseases but have not been studied as a risk factor for cow's milk allergy in Ukrainian children. The study of these polymorphisms, which are located on the 5q31-33 chromosome, will facilitate further studies of the connection between certain mutations and the development of cow's milk allergy. Epigenetic studies indicate that genes located on the 5q31-33 chromosome have a major influence on regulating basal serum immunoglobulin E (IgE) levels.

4. In the Methods Section, the Genotyping musty be better described (Linkage disequilibrium analysis).

Response: We thank Reviewer 1 for recommending this addition. We have added a better description of genotyping to the Methods section in the new subsection “2.5. Genotyping.

5. The MAF of these 2 SNPs could be reported.

Response: We thank Reviewer 1 for this comment. Although we know that the minor allele frequency (MAF) is widely used in population genetics, we calculated the major allele frequencies of the polymorphic variants of the IL-13 R130Q and IL-4 T589C genes and presented them in Table 1.

 6. The figures are too small and unclear.

Response: We thank Reviewer 1 for recommending that the figures be improved. We enlarged the figures and the numbers within them. We labeled the axes and added asterisks to show statistically significant comparisons.

7. Is there a correlation between serum levels of IL-4 and IL-5 and their respective polymorphisms?

Response: We thank Reviewer 1 for this interesting question. We did not analyze the level of IL-5. At the end of the Results section we have added a presentation of the correlations between serum levels of IL-4 and IL-13 and their respective polymorphisms.

 8. What is known about the function of these 2 polymorphisms on the gene expression?

Response: We thank Reviewer 1 for this interesting question.

Font was changed to Arial and bold and italics were removed.

Allergic disease represents a significant global health burden, and the disease incidence continues to rise in urban areas of the world. As such, a better understanding of the basic immune mechanisms underlying disease pathology is  likely to be the key to developing therapeutic interventions to both prevent disease onset as well as to ameliorate disease morbidity in those individuals already suffering from a disorder linked to type-2 inflammation. Two factors central to type-2 immunity are  IL-4 and IL-13. These two cytokines have been linked to virtually all of the major disease hallmarks associated with type-2 inflammation. Therefore, IL-4 and IL-13 and their regulatory pathways represent ideal targets for suppressing disease. However, despite sharing many common regulatory pathways and receptors, these cytokines perform very distinct functions during a type-2 immune response. Given the key role of IL-4 and IL-13 in type-2 inflammation, a significant amount of research has been performed to better understand the cellular and molecular mechanisms regulating IL-4 and IL-13 production. Based on their shared usage of lineage-determining factors STAT6 and GATA3, it is commonly believed  that IL-4 and IL-13 are coordinately expressed within immune cells [K. Bao, R. Lee Reinhardt. The differential expression of IL-4 and IL-13 and its impact on type-2 immunity. Cytokine 2016, 75(1): 25–37. doi: 10.1016/j.cyto.2015.05.008].

Reviewer 2 Report

It is known that gene polymorphism is important factor in predisposition to various diseases. The performed analysis were important in the aspect of understanding other genetic factors (like SNP) in the etiopathogenesis of allergy. The study of IL-4 and IL-13 polymorphisms is justified and necessary, as both cytokines are markers of inflammation directly related to food allergy.

The food allergy epidemic, especially to cow's milk, is a serious health and social problem. Therefore, the research carried out by the authors, and the purpose of the study are needed to develop knowledge about the pathomechanism of food allergy, and give a probable possibility of determining the predisposition to the development of hypersensitivity in children.

The paper contains the correct introduction and the research methodology has been well selected. The patient groups were correctly described, including the exclusions of the participants (section 2.1. Study Participants). I am a bit concerned about the number of patients participating in the study, but I understand that research work with biological material collected from young children (1-3 years old) has its limitations and requires special regulations.

The discussion is extensive and has a great value not only for scientists, but also for parents, doctors and therapists. I recommend that anyone who work with food allergy in children, should read this article. The estimation of protective genotypes and those predisposing to the development of food allergy may be an important tool in allergy prevention and disease monitoring.

COMMENT :

The results are described correctly, but please use the same font size in the manuscript and improve the quality and size of Figures 1 and 2.

Author Response

Response to Reviewer 2 

Dear Reviewer 2! Thank You for Your questions and recommendations!

The results are described correctly, but please use the same font size in the manuscript and improve the quality and size of Figures 1 and 2.

Response: We thank Reviewer 2 for these comments. We have improved the quality of both figures. We enlarged the figures and the numbers within them. We labeled the axes and added asterisks to show statistically significant comparisons.

Round 2

Reviewer 1 Report

The text has been improved and modified according to the suggestions.

The paper could be accepted for the pubblication.